# Social relationships and their associations with affective symptoms of women with breast cancer: A scoping review

Yesol Yang[1]*, Yufen Lin[2], Grace Oforiwa Sikapokoo[3], Se Hee Min[4], Nicole Caviness-Ashe[4], Jing Zhang[5], Leila Ledbetter[6], Timiya S. Nolan[5]

**1** Ohio State University Comprehensive Cancer Center-James, Columbus, Ohio, United States of America, **2** Nell Hodgson Woodruff School of Nursing, Emory University, Atlanta, Georgia, United States of America, **3** School of Communication Studies, Ohio University, Athens, Ohio, United States of America, **4** School of Nursing, Duke University, Durham, North Carolina, United States of America, **5** College of Nursing, Ohio State University, Columbus, Ohio, United States of America, **6** Duke University Medical Center Library, Durham, North Carolina, United States of America

* yang.6310@osu.edu

**Data Availability Statement:** All relevant data are within the manuscript and its Supporting Information files. Also, data is stored here: DOI 10. 17605/OSF.IO/F3ME9; https://osf.io/f3me9/.

## Abstract

### Background

Problems in affective and cognitive functioning are among the most common concurrent symptoms that breast cancer patients report. Social relationships may provide some explanations of the clinical variability in affective-cognitive symptoms. Evidence suggests that social relationships (functional and structural aspects) can be associated with patients' affective-cognitive symptoms; however, such an association has not been well studied in the context of breast cancer.

### Purpose

The purpose of this scoping review was to address the following question: What social relationships are associated with affective-cognitive symptoms of women with breast cancer?

### Methods

This scoping review used the framework proposed by Arksey and O'Malley and PRISMA-Sc. Studies published by February 2022 were searched using four databases: MEDLINE (PubMed), Embase (Elsevier), PsycINFO (EBSCOhost), and Web of Science (Clarivate). All retrieved citations were independently screened and eligibility for inclusion was determined by study team members. Extracted data included research aims, design, sample, type and measures of social relationships (functional and structural), and the association between social relationships and affective-cognitive symptoms

### Results

A total of 70 studies were included. Affective symptoms were positively associated with social support, family functioning, quality of relationships, social networks, and social integration, whereas the negative association was found with social constraints.

**Funding:** The author(s) received no specific funding for this work.

**Competing interests:** The authors have declared that no competing interests exist.

## Conclusion

Our findings suggest positive social relationships may mitigate affective symptoms of women with breast cancer. Thus, health care providers need to educate patients about the importance of building solid social relationships and encourage them to participate in a supportive network of friends and family members.

## Introduction

With advances in medical treatments, breast cancer mortality rates have steadily declined in recent years, resulting in an increase in 5-year survival rates [1]. According to the report from American Cancer Society in 2017, the overall survival rates have increased from 68% to 89% for White women and from 55% to 81% for Black women [2]. Resultantly, cancer is no longer viewed as an incurable acute disease. Instead, it follows the trajectories of chronic diseases that is characterized by periods of remission and exacerbation of symptoms [3]. Women with breast cancer often experience symptoms that co-occur (i.e., symptom clusters) during the disease trajectory [3]. For example, patients experience affective and cognitive problems (symptoms) concurrently. The co-occurrence of these symptoms is called a psychoneurological symptom cluster [4]. Further, these two symptoms within a psychoneurological cluster are strongly related to each other [4, 5].

Problems in affective and cognitive functioning are among the most common concurrent symptoms that breast cancer patients report [4, 5]. Affective symptoms include any mood disturbances that occur throughout the illness trajectory of cancer (e.g., depression, stress, anxiety, and fear), and such symptoms confers the risk of development of problems in memory, concentration, processing speed, and language (i.e., cognitive symptoms) [4, 5]. Severe affective-cognitive symptoms may result in poor adherence of cancer treatments [6] and lower levels of functioning status and quality of life [7]. For this reason, it is critical to identify who may be at risk for affective as well as cognitive symptoms.

Factors that contribute to affective-cognitive (i.e., psychoneurological) symptoms were identified as stress, hypothalamic-pituitary-adrenocortical axis dysfunction, cytokine dysregulation, telomere shortening, or DNA damage [4]; however, these factors do not sufficiently explain the variability in these symptoms. For example, some patients have reported persistent and high levels of mood disturbance and cognitive impairment for several years or more following cancer treatment [8]. This finding suggests a need to investigate other potential factors that can explain their clinical variabilities.

Social relationships may provide some explanations of the clinical variability in affective-cognitive symptoms. Social relationships refer to the connections between individuals that they perceive to have personal meaning [9]. These relationships can be characterized as aspects, structural and functional. The structural aspect reflects the size, scope, and connectedness of social relations (e.g., social integration, social network), while the functional aspect covers the interpersonal interaction within the structure of the social relations (e.g., social support, family cohesion) [9]. Although the exact underlying mechanism of the association between social relationships and affective-cognitive symptoms remains unknown, several studies have examined this association. Recent systematic reviews have reported that the older populations showed a greater decline in their cognition when their social relationship was functionally and structurally poor [10, 11]. Other studies have found that patients who had greater social support and cohesive relationships with their family members showed fewer

depressive symptoms [12, 13]. Specifically, breast cancer survivors demonstrated higher levels of depressive symptoms over the trajectory of their illness when they received lower levels of social support [14].

Taken together, greater social relationships (both functional and structural aspects) appear to be associated with fewer affective-cognitive symptoms in breast cancer patients. However, there is no comprehensive understanding on whether or which social relationships characteristics relate to BC patients' affective-cognitive symptoms. Therefore, the understanding of social relationship characteristics associated with affective-cognitive symptom in breast cancer patients may be important to as a basis for development of prevention and interventions to manage those symptoms.

## Purpose

The purpose of this scoping review was to map the literature that has investigated both aspects of social relationships (functional and structural) and their links to affective-cognitive symptoms in breast cancer patients. This review paper addresses the following question: What social relationships are associated with affective-cognitive symptoms of women with breast cancer? This will lay the foundation for studies that explicate the mechanism of affective-cognitive symptoms in breast cancer patients. This understanding will also allow clinicians to identify patients more precisely at risk for affective-cognitive symptoms associated with social relationships and will contribute to the development of strategies to prevent and manage these symptoms.

## Materials and methods

We reported the findings using the five methodological stages of scoping review developed by Arksey and O'Malley [15]. This review was conducted based on the following stages: 1) identifying the research question, 2) identifying relevant studies, 3) selecting studies, 4) charting the data, and 5) collating, summarizing, and reporting the results. A scoping review approaches was used because it helps clarify key concept related to outcomes as well as identify the types of available evidence [16]. We reported using the Preferred Reporting Items for Systematic Reviews and Meta-Analyses extension for Scoping Reviews (PRISMA-ScR) [17].

### Stage 1. Identifying research questions

We applied the PCO model to develop our research question [15]: "What social relationships ("C", concept) are associated with affective-cognitive symptoms ("O", outcome) of women with breast cancer ("P", population)?" We limited our study population of women with breast cancer aged 18 years and above because of different trajectories and manifestations of cognitive symptoms that children with cancer show compared with adults [18]. Table 1 describes eligibility criteria for the studies that were included in this scoping review.

### Stage 2. Identifying relevant studies

We developed relevant search terms in collaboration with a librarian included a mix of keywords and database specific subject headings representing women, breast cancer, affective symptoms and social relationships. The search was translated and conducted by a medical research librarian on February 15, 2022 using four databases: MEDLINE (PubMed), Embase (Elsevier), PsycINFO (EBSCOhost), and Web of Science (Clarivate). Editorials, letters, and comments were excluded, as were animal-only studies and studies involving pediatric populations. Reproducible search strategies for all databases can be found in S1 Table. We reviewed

**Table 1. Inclusion criteria and exclusion criteria.**

| Study characteristics | Inclusion criteria | Exclusion criteria |
|---|---|---|
| Study types | • Observational or experimental studies<br>• Peer-reviewed journals<br>• Published in English by February, 2022 | • Qualitative, case reports, editorial, letters, comments, doctoral dissertation, and conference proceeding |
| Populations | • Women with breast cancer<br>• Aged 18 years or older<br>• Undergoing active or completed cancer treatments | • Animal<br>• Adolescence<br>• Child<br>• Infant |
| Social relationships | • Interactions, connections, and relationships between individuals (e.g., social support, social network, social integration, social ties, relationships with family, caregivers, neighborhoods, and co-workers) | • Social contacts and interactions that are fleeting, incidental, or perceived to have limited significances (e.g., retail employees, time-limited interaction with service providers) |
| Affective symptoms | • Mood disturbances or fluctuating affective states (e.g., anxiety, depression, mood/psychological disturbances | • Mood disturbances attributed to non-cancer causes such as psychiatric illness. |
| Cognitive symptoms | • Cognitive impairments or decline | • Cognitive deficits attributed to non-cancer causes such as neurological illness, dementia, stroke, brain injury or delirium |
| Association between social relationships and affective-cognitive symptoms | | • Studies that did not assess the association between social relationships and patients' affective-cognitive symptom |

the results for existing review articles and determined that no review articles currently exist on our topic.

## Stage 3. Study selection

The search identified a total of 5,386 references that were imported into Covidence, a systematic review screening tool (Covidence systematic review software, Veritas Health Innovation, Melbourne, Australia. Available at www.covidence.org). Duplicate citations (n = 1,504) were automatically identified and removed by Covidence. The software ensures that two reviewers independently screened a total of 3,882 references by title and abstract. Studies were excluded if they did not clearly meet inclusion criteria, and of those, 3,723 references were deemed irrelevant and excluded. Upon the completion of screening titles and abstracts, any disagreements were resolved by discussion. One hundred fifty-nine citations were identified for full text assessment. At the full text review stage, articles were independently read by two different members of team (YY, YL, GS). During the full-text review, each study was reviewed independently to determine the final sample. Full-text studies that did not meet the inclusion criteria were excluded, and the reasons for exclusion were noted. Disagreement between the team members were resolved through discussion. Seventy articles were confirmed to be included in the final set for data extraction.

## Stage 4. Charting the data

Our team developed a data extraction tool and determined which data should be extracted from studies to answer the research question. Two team members (YY, GS) independently piloted data abstraction from the first fifteen included studies using the data charting form. Then, they discussed the process and their results to confirm whether their approaches to data extraction were consistent. Questions arising when piloting the extract data form were discussed with the other team members (YL, TN). After piloting the form, two team members (YY, GS) independently recorded the following data from selected studies on the data charting form: 1) authors, 2) country of study, 3) year of publication, 4) study design, 5) sample characteristics (sample size, age, and type of cancer treatment), 6) type of social relationships 7)

affective-cognitive symptoms and measurements, and 8) key findings (the association between social relationships and affective-cognitive symptoms).

## Stage 5. Collating, summarizing and reporting the results

Our team collated, summarized, and reported all data obtained in stage 4 to map the knowledge on social relationships associated with affective-cognitive symptoms of adult women with breast cancer. The studies in the final sample were tabulated based on social relationships (e.g., functional or structural aspect of social relations). A table for the final sample was created and included the information on authors, years of publication, country of study, study population, type of social relationships, measures of affective-cognitive symptoms, and the association between social relationships and affective-cognitive symptoms. Verification of data accuracy was impudently conducted by six research team members (YY, YL, GS, SM, NC, JZ).

## Results

### Study characteristics

Table 2 includes sixty-five studies that met the inclusion criteria. Fig 1 presents study selection by flowchart as per PRISMA guidelines. The reviewed studies were conducted in 22 countries with majority conducted in the US (n = 30) and Canada (n = 4). Of 70 studies, 36 were cross-sectional, 4 were randomized controlled trials (RCTs), 9 were longitudinal, and 21 were a secondary analysis from a cross-sectional, longitudinal, or multiple-institutional cohort study. Two studies included this review used the same dataset [19, 20]. The sample size of dyadic studies (included both patients and their spouses/partners/family caregivers) ranging from 92 to 470, and the sample size of the remaining 60 non-dyadic studies ranged from 25 to 2235 patients. The mean age of patients who participated in this study ranged from 36.7 to 66.7 years old. Also, participants in the published studies from the US were White, followed by Black (African American), Latina or Asian. Of the included studies, three dealt with patients living with metastatic/advanced breast cancer. Additionally, cancer treatments that patients received were varied including chemotherapy, surgery, hormone, radiation, and targeted therapy.

### Association between social relationships and affective symptoms

In this review, social relationships were classified as functional and structural aspects of social relations. Of the included 70 studies, 64 focused on functional aspects of social relationships, and the remaining 6 reported on structural aspects of social relationships. Interestingly, none of the included 70 studies examined the association between social relationships and cognitive symptoms of breast cancer patients; thus, in this paper, we focused only on the affective symptoms of breast cancer patients and their association with patients' social relationships (Table 3).

### Functional aspects of social relationships

Social support, satisfaction of social support, quality of the relationship, social constraints, and family functioning (including family conflict and family stress) are functional social relationships included in this review.

**Social support.** Fifty-three studies examined the association between social support and affective symptoms among breast cancer patients. Of those 53 studies, four did not find any associations of affective symptoms with social support [21–24], whereas one showed that patients' affective symptoms can be changed depending on the source of provided support was

**Table 2. Study characteristics.**

| Author (year), Country | Study design | Sample characteristics | | | |
|---|---|---|---|---|---|
| | | N | Race/ethnicity | Age (mean, SD) | Tx |
| Roberts et al.,1994 (USA) | Secondary analysis | 135 women with breast cancer | NR | 56.2 (SD = 11.9) | Surgery (100%) |
| Neuling et al., 1988 (Australia) | Longitudinal | 58 women with breast cancer | NR | Median = 54 | Surgery (100%) |
| Koopman et al., 1998 (USA) | Cross-sectional | 102 women with metastatic and/or recurrent breast cancer | White (88.2%); Asian-American (4.9%); African-American (1%); Hispanic/Latina (2%); Native American (2%); Other (2%) | 53.1 (SD = 10.8) | *CTx* (52%); *Hormone* (76.5%) |
| Lee et al., 2004 (Korea) | Cross-sectional | 134 women receiving chemotherapy for breast cancer | Korean (100%) | 45.29 (SD = 8.75) | *CTx* (100%) |
| Maly et al., 2005 (USA) | Cross-sectional | 222 women with newly diagnosed breast cancer | White (64%); African-American (12%); Latina (23%); Other (1%) | 66.7 (SD = 7.9) | *Surgery* (31.5%)*; RTx* (40%); *CTx* (37.4%) |
| Palesh et al., 2006 (USA) | Cross-sectional | 82 women recently diagnosed with breast cancer stage 0-III | NR | 57.4 (SD = 11.5) | *Surgery* (mastectomy, 43%; lumpectomy, 79.3%); *CTx* (50%); *RTx* (59.8%); *Hormone* (42.7%) |
| Friedman et al., 2006 (USA) | Cross-sectional | 81 women with breast cancer | African-American; Hispanic; White (% NR) | 52 (SD = 10.2) | Surgery (74%); CTx (88.9%) |
| Porter et al., 2006 (USA) | Secondary analysis | 524 women with breast cancer | White (70.6%); African-American (29.4%) | 64.5 (SD = 8.9) | *Surgery* (98.6%); *CTx* (23%); *RTx* (27%); *Hormone* (28%) |
| Kim & Morrow, 2007 (USA) | Secondary analysis | 539 women with breast cancer | White (94%) | 51 | *CTx* (100%) |
| Nausheen & Kamal, 2007 (Pakistan) | Cross-sectional | 82 Pakistani women with breast cancer | Pakistan (100%) | 42.5 | *Surgery* (90%) |
| Von Ah & Kang, 2008 (USA) | Longitudinal | 49 American women with newly breast cancer stage 0-III | White (61%); African-American (29%); Asian-American (4%); Hispanic-American (2%); Native American (4%) | 52.3 (SD = 9.6) | *CTx+RTx* (51%) |
| Gellaitry et al., 2010 (UK) | RCT | 80 women with breast cancer (n = 38 intervention group) | NR | 58.4 (SD = 10.8) | *Surgery* (100%); *CTx* (53%); *RTx* (100%); *Hormone* (82%) |
| Gorman et al., 2010 (USA) | Cross-sectional | 131 women with early-stage breast cancer | White (87.8%); Other (12.2%) | 36.7 (at diagnosis) | *CTx* (88.6%); *RTx* (55.7%) |
| Hasson-Ohayon et al., 2010 (Israel) | Cross-sectional | 150 dyads of women with breast cancer stage III-IV and their spouses | Israel (100%) | Patients: 53.15 (SD = 10.28) | Mostly not on active treatment |
| Kim et al., 2010 (USA) | Cross-sectional | 231 undeserved women with breast cancer | White (62.3%); African-American (35.9%); other minorities (1.7%) | 51 | NR |
| Talley et al., 2010 (USA) | Secondary analysis | 163 women with breast cancer | White (94.5%); Black (2.5%); Other (1.8%) | 57.33 (SD = 11.22) | NR |
| Cohen et al., 2011 (Israel) | Cross-sectional | 56 women with breast cancer (stage I-III) | Arabs (100%) | 50.6 (SD = 8.7) | NR |
| Hill et al., 2011 (UK) | Longitudinal | 260 women with breast cancer | NR | 151 patients aged 51–64 years | *Surgery* (100%) |
| Lee et al., 2011 (Korea) | Secondary analysis | 286 women with breast cancer stage I-III | Korean (100%) | 47 (SD = 10) | *Breast Conserving Surgery* (82.5%); *Mastectomy* (16.4%); *CTx* (86.7%); *RTx* (82.5%); *Hormone* (82.2%) |
| Liu et al., 2011 (China) | Cross-sectional | 401 women with breast cancer | Chinese (100%) | 46.9 (SD = 10.1) | NR |

*(Continued)*

**Table 2.** (Continued)

| Author (year), Country | Study design | Sample characteristics | | | |
|---|---|---|---|---|---|
| | | N | Race/ethnicity | Age (mean, SD) | Tx |
| Boinon et al., 2012 (France) | Cross-sectional | 113 women with breast cancer | NR | 52.8 (SD = 10.17) | *Surgery (31%)* |
| Jones et al., 2012 (Canada) | Cross-sectional | 131 women with early-stage breast cancer | NR | 54.6 (SD = 9.13) | *Surgery (96.9%) CTx (57.3%); RT (51.1%)* |
| Mallinckrodt et al., 2012 (USA) | Longitudinal | 154 women with breast cancer | White non-Hispanic (97%); African-American (2.6%); Hispanic (0.6%) | 58.97 (SD = 12.33) | CTx+RTx (19%); CTx (32%); RTx (18%) |
| Popoola & Adewuya, 2012 (Nigeria) | Cross-sectional | 124 women with breast cancer | Nigerian (100%) | NR | Surgery (9.7%) Surgery+CTx (39.5%); Surgery+CTx +RTx (50.8%) |
| So et al., 2013 (China) | Secondary analysis | 279 women with breast cancer | Chinese (100%) | NR | NR |
| Waters et al., 2013 (USA) | Secondary analysis | 480 women with breast cancer stage 0-IIA | White (81.5%); non-White (18.5%) | 58.3 (SD = 10.6) | *Surgery (100%); CTx (24.8%); RTx (60%);* Hormone (51.3%) |
| Yi & Kim, 2013 (Korea) | Cross-sectional | 258 Korean women with breast cancer | Korean (100%) | 47.45 (SD = 7.37) | *Surgery (98.8%); CTx (83.3%); RTx (55.4%); Hormone (55%)* |
| Boinon et al., 2014 (France) | Longitudinal | 102 women with breast cancer | French (100%) | 52.9 (SD = 10.2) | *Surgery (31.4%); CTx+RTx (100%)* |
| Hasson-Ohayon et al., 2014 (Israel) | Secondary analysis | 150 women with advanced breast cancer | Israel (100%) | Younger: 45.67 (SD = 6.55) Older: 62.16 (SD = 5.70) | NR |
| Hughes et al., 2014 (USA) | Longitudinal | 164 women with breast cancer stage 0-IIIA | White (80.5%); Black (12.8%); Other (6.7%) | 56.13 (SD = 11.47) | *Surgery (32.9%); Surgery +RTx(27.4%); Surgery+CTx (15.2%); Surgery+RTx+CTx (23.8%)* |
| Schleife et al., 2014 (Germany) | Secondary analysis | 107 women with breast cancer | NR | 56.4 (SD = 10.5) | *Surgery (96%); CTx (98%)* |
| Wang et al., 2014 (China) | Cross-sectional | 123 women with breast cancer | Chinese (100%) | 49.7 (SD = 9.6) | NR |
| Borstelmann et al., 2015 (USA) | Secondary analysis | 675 women with breast cancer stage I-III | White (86%); Non-white (14%) | 35.4 | *Surgery (84%); CTx (76%)* |
| Ozkaraman et al., 2015 (Turkey) | Cross-sectional | 128 breast cancer patients | NR | 51.13 (SD = 8.48) | NR |
| Alfonsson et al., 2016 (Sweden) | Longitudinal | 833 women with breast cancer | Sweden (100%) | 60.6 (SD = 11.6) | *CTx (38%); Target-drug (8%)* |
| Malicka et al., 2016 (Poland) | Cross-sectional | 25 women with breast cancer | Polish (100%) | 63.2 (SD = 7.0) | *Surgery (100%)* |
| Berhili et al., 2017 (Morocco) | Cross-sectional | 446 women with breast cancer | NR | 50 (SD = 8) | *Surgery (21%); CTx (38%) RTx (17%); Hormone (33%)* |
| Fong et al., 2017 (Canada) | Secondary analysis | 157 women with breast cancer | White (85%) | 55 (SD = 11) | *Lumpectomy (60.1%); Mastectomy (57.8%); CTx (63.6%); RTx (85%); Hormone (52.6%)* |
| Moon et al., 2017 (USA) | Secondary analysis | 661 women with newly diagnosed with breast cancer | White (89%); Minority (9.8%); Not applicable (1.3%) | 51.18 (SD = 9.05) | NR |

(*Continued*)

**Table 2.** (Continued)

| Author (year), Country | Study design | Sample characteristics | | | |
|---|---|---|---|---|---|
| | | N | Race/ethnicity | Age (mean, SD) | Tx |
| Schellekens et al., 2017 (Canada) | RCT | 139 women with breast cancer stage I-III (MBCR; n = 69 and SET; n = 70) | Canadian (100%) | MBCR: 54.9 (SD = 9.2) SET: 53.2 (SD = 9.8) | NR |
| Su et al., 2017 (Taiwan) | Cross-sectional | 300 women with breast cancer | Taiwanese (100%) | 48.16 (SD = 9.07) | *RTx* (58.7%);*CTx* (71.7%) *Hormone* (70%); *Target-drug* (22.7%) |
| Thompson et al., 2017 (USA) | Secondary analysis | 227 African American women with breast cancer | African American (100%) | 56 (SD = 10) | *Surgery* (68.8%); *CTx* (49.6%) *RTx* (77.4%); *Hormone* (63.1%) |
| Tomita et al., 2017 (Japan) | Secondary analysis | 157 women with breast cancer | Japanese (100%) | 59.08 (SD = 10.06) | *Surgery* (94.9%);*CTx* (50.3%); *RTx* (65.6%);*Hormone* (75.8%) |
| Bright & Stanton, 2018 (USA) | Longitudinal | 130 women with breast cancer | White (73.1%); Asian (9.2%); Latina (8.5%); African American (3.1%); Native American/Alaskan Native (0.8%); Other (5.4%) | 54.2 (SD = 11.7) | *Surgery* (99.2%); *Hormone* (92.3%) |
| Schmidt et al., 2018 (Germany) | Secondary analysis | 225 women with breast cancer | Germany (100%) | 54.3 (SD = 9.5) | *CTx* (37.6%) |
| Escalera et al., 2019 (USA) | Secondary analysis | 151 Latinas with breast cancer stage 0-IIIc | NR | 50.5 (SD = 10.9) | *Surgery* (100%);*CTx* (16.6%); *RTx* (27.8%); *CTx+RTx* (39.7%) |
| Wondimagegnehu et al., 2019 (Ethiopia) | Cross-sectional | 428 women with breast cancer | Ethiopian (100%) | Median = 40 | NR |
| Janowski et al., 2020 (Poland) | Cross-sectional | 70 women with breast cancer | Polish (100%) | 56.52 (SD = 14.18) | *Surgery* (100%) |
| Schmidt & Andrykowski 2004 (USA) | Cross-sectional | 210 women with breast cancer | White (91%); African-American (1.4%); Asian (1%); Latino/Hispanic (1%); Native American (0.5%); Other (4.3%) | 47.4 (SD = 8.4) | *Surgery* (88.1%); *CTx* (26.2%); *RTx* (12.9%); *CTx+RTx* (52.9%) |
| Wong et al., 2018 (USA) | Cross-sectional | 96 Chinese American breast cancer survivors | Chinese-American (100%) | 54.54 (SD = 7.91) | NR |
| Lally et al., 2019 (USA) | RCT | 100 women within 0–2 months of first, stage 0-II breast cancer survivors | White (93%); African-American (3%); American-Indian (1%); Asian (1%) | 54.2 (SD = 9.9) | NR |
| Lueboonthavatchai, 2007 (Thailand) | Cross-sectional | 300 women with breast cancer | NR | NR | 50.09 (SD = 11.01) |
| Mantani et al., 2007 (Japan) | Cross-sectional | 46 women with breast cancer stage I or II and their husbands | Japanese (100%) | Patients: 52.3 (SD = 10.5) | *Surgery* (100%) *CTx,RTx, hormone* (87%) |
| Ashing-Giwa et al., 2013 (USA) | Secondary analysis | 232 women with Latina breast cancer stage 0-III | Mexican (73%); Central-American (13%); South-American (9%); US-born Latinas (5%) | 53 (SD = 10.6) | *Surgery* (95%); *CTx*(70%); *RTx* (70%); *Hormone* (66%) |
| Segrin et al., 2018 (USA) | Cross-sectional | 230 dyads of Latinas with breast cancer and their family caregivers | White (85%); Hispanic (14%); Other (1%) | Patients: 50.19 (SD = 10.4) Caregivers: 44.20 (SD = 13.2) | *Surgery* (60%); *CTx* (82.6%) *RTx* (27%); *Hormone* (14.8%) |
| Aguado Loi et al., 2013 (USA) | Secondary analysis | 68 Latinas diagnosed with breast cancer | Latino/Hispanic (100%); | 55.4 (SD = 10.4) | *Surgery* (95.6%); *CTx* (63.2%); *RTx* (48.5%); *Hormone* (69.1%) |

(*Continued*)

**Table 2.** (Continued)

| Author (year), Country | Study design | Sample characteristics | | | |
|---|---|---|---|---|---|
| | | N | Race/ethnicity | Age (mean, SD) | Tx |
| Giese-Davis & Hermanson, 2000 (USA) | Cross-sectional | 125 women with metastatic breast cancer | White (87%); Asian-American (6%); Hispanic-Latina (2%); Native American (2%); African-American (1%); Other (2%) | 53 (SD = 10.7) | NR |
| Manne et al., 2007 (USA) | Secondary analysis | 235 women with breast cancer and their significant others | White (patients: 89% and partners: 91%) | 50 (SD = 9.9) | *Surgery* (100%); *CTx* (75%); *RTx* (13%) |
| Segrin et al., 2007 (USA) | Secondary analysis | 96 dyads of women with breast cancer stage I-III and their partners | White (85%); Hispanic (14%); Other (1%) | Patients: 54.11 (SD = 10.6) Partners: 51.68 (SD = 14.8) | *CTx* (75%); *RTx* (54%); *Hormone* (36%) |
| Al-Zaben et al., 2015 (Saudi Arabia) | Cross-sectional | 49 married women with breast cancer | Arabs (100%) | 48.9 (SD = 7.1) | *Surgery* (89.8%); CTx (83.7%); RTx (57.1%) |
| Simpson et al., 2002 (Canada) | RCT | 89 women with breast cancer | NR | 49.3 (SD = 7.7) | NR |
| Brothers & Andersen, 2009 (USA) | Longitudinal | 67 women with breast cancer | White (93%); African-American (7%) | 54 (SD = 11) | Surgery (28%); *CTx* (43%); *RTx* (19%); *Hormone* (39%) |
| Gagliardi et al., 2009 (Italy) | Cross-sectional | 47 women with breast cancer at low or intermediate high risk | Italian (100%) | 54.28 (SD = 8.4) | *Surgery* (100%) |
| Puigpinos-Riera et al., 2018 (Spain) | Secondary analysis | 2235 women with breast cancer | Spanish (100%) | NR | NR |
| Wang et al., 2019 (USA) | Cross-sectional | 436 women with breast cancer stage 0-III | Chinese (100%) | 21–50 yrs (27.52%), 51–64 yrs (48.17%), 65 or older yrs (24.31%) | NR |
| Debretsova &Derakshan., 2021 (UK) | Cross-sectional | 59 women with breast cancer stage IV | NR | 49.97 (SD = 9.17) | *Surgery* (80%); CTx (34%); RTx(10%); Hormone(71%) |
| Fisher et al., 2021 (Germany) | Cross-sectional | 327 women with breast cancer stage I-III | White (62.1%); Black (29.7%); Two or more races (2.8%); Asian (2.8%); American Indian or Alaskan Native (0.3%); NR (1.5%) | 57.19 (SD = 11.87) | Surgery (100%); CTx (8.3%); RTx (10.8%) |
| Liu et al., 2021 (China) | Cross-sectional | 389 women with breast cancer | Chinese (100%) | ≤35yrs(11.3%); 35-50 (43.7%);50-65 (39.3%);>65(5.7%) | NR |
| Zamanian et al., 2021 (Iran) | Cross-sectional | 223 women with breast cancer | Persian (100%) | 47.14 (SD = 9.13) | Surgery (69.7%) |
| Okati-Aliabad et al., 2022 (Iran) | Cross-sectional | 120 Women with breast cancer stage I-IV | Persian (100%) | 47.35 (SD = 10.67) | Surgery (87.5%); CTx (94.2%); RTx (64.2%) |

*NR = Not reported; CTx = Chemotherapy; RTx = Radiation therapy

[21]. A reduction in patients' depression was reported when patients received peer support from patients who are newly diagnosed with cancer rather than from patients who are undergoing active treatment [21].

Among 49 studies that reported significant association with affective symptoms, 37 investigated the association of patients' affective symptoms with the quantity of social support that patients received. The quantity of social support refers to the amount of social support that is available to patients (e.g., frequency of meetings) [25]. Patients showed lower levels of anxiety, depression, worry, mood disturbances, and psychological/mental distress when they received a

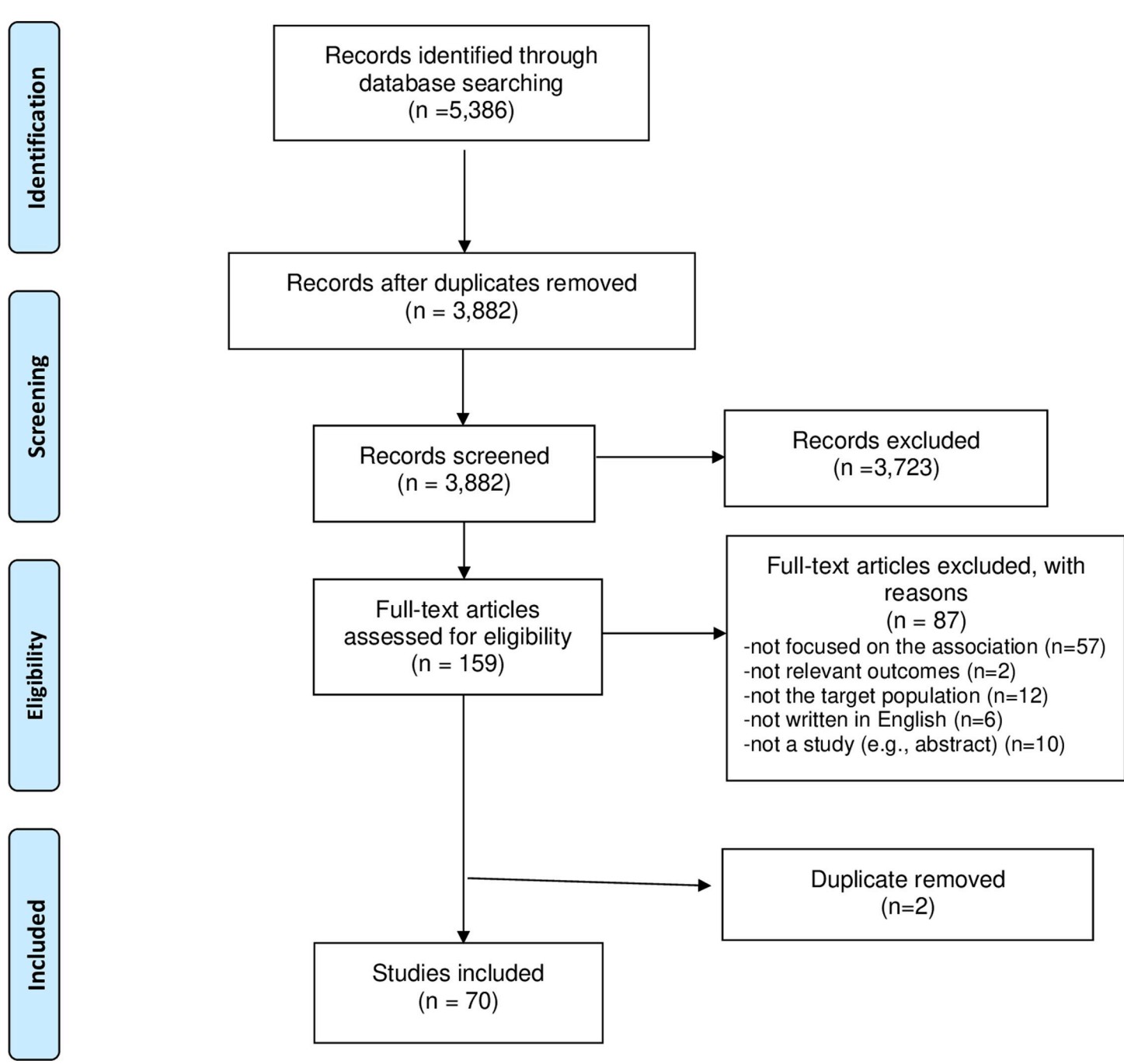

**Fig 1. PRISMA-ScR flow diagram for the study selection process.**

greater quantity of social support [14, 20, 24, 26–50]. Furthermore, some studies reported that the quantity of social support can predict the levels of patients' affective symptoms including their emotional well-being [51–58].

In addition to the quantity of social support, seven studies reported an association between type of social support and affective symptoms. Emotional (i.e., subjective) support, defined as support that includes the provision of care, empathy, and trust, was found to be most helpful to decrease patients' depression and anxiety.[37, 47, 56, 59, 60] In other words, as patients

**Table 3. Characteristics of studies regarding social relationships associated with the patient's affective symptoms.**

| Author (year) | Social relationship (measures) | Affective symptoms (measures) | Analysis adjusts for | Key findings |
|---|---|---|---|---|
| | | | | **Functional Aspect of Social Relationships** |
| **Social support** | | | | |
| Roberts et al.,1994 | Social support (SSQ) | Psychological status (SCL-90-R) & (distress GSI) | Desirability | 1) Single patients who had support from friend demonstrated lower depression (r = -0.44), anxiety (r = -0.38), and overall severity of psychological distress (r = -0.41) (all p <0.05). 2) Married patients who had support from spouse demonstrated lower depression (r = -0.27) as well as overall severity of psychological distress (r = -0.27) (both p<0.01). |
| Neuling et al., 1988 | Social support (MDSS) | Anxiety (STAI) Depression (Wakefield Self-Assessment Depression Inventory) | NR | *In hospital*, 1) Anxiety was related to amount of support received from friends [F(1,49) = 5.84;p<0.05] and satisfaction with support from family members [F(1,50) = 4.54, p<0.05]. 2) Depression was related to the amount of support from friends [F(1,49) = 6.50, p<0.05] and satisfaction with family support [F(1,50) = 5.58, p<0.05]. *1-month post operation*, Anxiety was related to the amount of support received from friends [F(1,37) = 6.77, p<0.05] *3-month post operation*, 1) Anxiety was not related to the amount of support but was related to satisfaction with support from family [F(1,34) = 9.72, p<0.005] 2) Depression was related to satisfaction with support from family [F(1,34) = 5.60, p<0.05] |
| Koopman et al., 1998 | Social support (Yale Social Support Index & single item measure) | Mood disturbance (POMS) | NR | Patients' mood disturbances were positively associated with aversive social support. |
| Lee et al., 2004 | Social Support (SSS) | Mood disturbance (Linear Analogue Self-Assessment Scale) | NR | Patients with low social support reported higher mood disturbance (r = -0.25, p = 0.004). |
| Maly et al., 2005 | Emotional &Instrumental support (items developed based on qualitative interview) | Depression (CES-D) Anxiety (STAI-S) | Socio-demographics, cancer stage, treatment type, comorbidity | 1) Patients demonstrated lower depressive when they had partners who helped around the house (β = -0.16, *p* = 0.048). 2) In white women (patients), they showed more anxiety when they had other family members or friends who helped with bathing or dressing (β = 0.20, *p* = 0.028) 3) In non-white women (patients), their anxiety became lower when they had children who listened to concerns or worries (β = -0.30, *p* = 0.044) and helped around the house (β = -0.25, *p* = 0.046). 4) In non-white women (patients), their depression became lower when they had children who helped around the house (β = -0.30, *p* = 0.02). |
| Palesh et al., 2006 | Social support (UCLA Social Support Inventory) | Mood disturbance (POMS) | NR | No relationship was found between mood disturbances and satisfaction with social support |
| Porter et al., 2006 | Social support satisfaction (SSQ) | Negative mood (POMS-SF) | NR | Patients demonstrated less negative mood state when their satisfaction with social support increased (β = -0.087, *t* = -2.041). |
| Friedman et al., 2006 | Social support (SSQ) | Mood disturbances (TMD and POMS-SF) | NR | No association was found between mood disturbances and social support |
| Kim & Morrow, 2007 | Family support (FES) | Anxiety (STAI) | Emetic score | Higher family support predicted lower patients' anxiety level (β = -0.36, *p*<0.001). |

*(Continued)*

**Table 3.** (Continued)

| Author (year) | Social relationship (measures) | Affective symptoms (measures) | Analysis adjusts for | Key findings |
|---|---|---|---|---|
| Nausheen & Kamal, 2007 | Familial social support (FSSS) | Depression (SSDS) | NR | 1) Patients showed less depression when they had strong familiar support (r = -0.85, $p<0.001$) |
| Von Ah & Kang, 2008 | Emotional and aid support (NSSQ) | Mood disturbance (POMS-SF) | NR | 1) Emotional support was associated with mood disturbance before (r = -.34, $p < .01$), after (r = -.47, $p < .001$) adjuvant therapy. 2) Aid support was associated with mood disturbances during (r = -.38, $p < .001$) adjuvant therapy. 3) Prior to adjuvant therapy, aid support had indirect effect on mood disturbance whereas emotional support has both indirect (ß = -0.42, $p < .05$) and direct effect (ß = -0.53) on mood disturbances 4) During adjuvant therapy, aid support has indirect effect of mood disturbance 5) After adjuvant therapy, emotional support has both direct (ß = -0.39, $p<0.01$) and indirect effects on mood disturbances. |
| Gellaitry et al., 2010 | Social Support (Significant Others Scale) | Psychological well-being (POMS) | Baseline measures | In intervention group, patients demonstrated less depression when they were satisfied with emotional support ($p<0.05$) |
| Gorman et al., 2010 | Social support (MOS-SSS) | Depressive symptoms (CES-D) | Demographic and clinical characteristics, randomized assignment | Patients with greater social support showed lower depressive symptoms ($p<0.0001$) |
| Hasson-Ohayon et al., 2010 | Agent of Support and Type of Support (CPASS) | Psychological distress (BSI) | NR | 1) Patients demonstrated lower depression when they had support from spouse (r = -0.16, $p<0.05$), family (r = -0.28, $p<0.01$), and friends (r = -0.24, $p<0.01$). 2) Patients demonstrated lower anxiety when they had support from family (r = -0.22, $p<0.01$). 3) Higher family support predicted lower patient's psychological distress (β = -0.32, $p<0.005$). 4) Higher family support predicted lower depression and anxiety (β = -0.20, $p<0.0057$; β = -0.19, $p<0.052$). |
| Kim et al., 2010 | Social support (developed from previous studies using six items) | Emotional well-being (FACT-B) | Age, education level, race, living status, cancer stage | 1) Patients with strong social support reported good emotional well-being (r = .34, $p < .001$) 2) Social support influenced emotional well-being (ß = 0.23, $p < .001$) |
| Talley et al., 2010 | Partner social support (Items developed by Alferi et al, 2001) | Depression (CES-D) | Age, income co-morbid illness, co-residence, distance from radiation treatment center, level of physical symptoms | Patients showed lower levels of depression when they had greater partner emotional support (β = -0.23, $p<0.05$). |
| Cohen et al., 2010 | Perceived social support (self-report 10-item) | Emotional distress (BSI-18) | Demographics | Perceived social support predicted the variance of emotional distress (β = -0.30, $p<0.01$) |
| Hill et al., 2011 | Perceived emotional support (MOS-SSS) | MD (Major depression) GAD (Generalized anxiety disorder) | History of psychiatric disorder | 1) During one year after cancer diagnosis, low social support predicted onset of MD (OR = 2.20, 95% CI 1.12–4.33, $p<0.05$)and GAD (OR = 2.51, 95% CI 1.05–5.97, $p<0.03$) 2) Low social support predicted the risk of onset of both MD (OR = 3.43, 95% CI 1.32–8.87, $p = 0.01$) and GAD (OR = 4.00, 95% CI 1.42–11.30, $p = 0.01$) |
| Lee et al., 2011 | Perceived Social Support (MOS-SSS) | Depressive Mood (SDS) | Demographics, menopausal status, BMI, exercise, drinking status. | Worsen emotional support ($p<0.001$), informational ($p = 0.04$) were associated with deteriorated depressive mood. |

(*Continued*)

**Table 3.** (Continued)

| Author (year) | Social relationship (measures) | Affective symptoms (measures) | Analysis adjusts for | Key findings |
|---|---|---|---|---|
| Liu et al., 2011 | Social support (social support rating scale) | Anxiety and depression (HADS) Psychological stress (Psychological stress scale) | NR | 1) Patients' psychological stress were associated with social support (subjective and objective) and its utility (all p<0.05) 2) Both anxiety (r = -0.196, p<0.01; r = -0.128, p<0.05) and depression (r = -0.141, p<0.01; r = -0.168, p<0.01) were associated with objective support and its utility. 3) Depression was associated with subjective social support (r = -0.315, p<0.01). 4) Subjective social support (β = -0.108, *p*<0.05) and its utility (β = 0.329, *p*<0.05) were predictors of anxiety. 5) Objective social support (β = -0.249, *p*<0.05) was a predictor of depression |
| Boinon et al., 2012 | Perceived social support (Cancer-specific questionnaire of social support) | Depressive symptom (BDI-SF) Negative affect (PANAS) | Demographics, time since surgery, social sharing variables | Patients with higher perceived negative support demonstrated higher depressive symptoms and negative affect (ß = 0.24, *p*<0.05; ß = 0,26, *p*<0.01) |
| Jones et al., 2012 | Social support (MSPSS) Unsupportive social interactions (USII) | Health anxiety (MIHT) Anxiety and depression (HADS) | Demographics, cancer-related variables, general anxiety and depression | 1) Patients had a tendency to worry about their health (health anxiety-affective dimension) when they had unsupportive social interactions (r = 0.36, p<0.001) 2) Patient's health anxiety-affective dimension was predicted by unsupportive interactions (β = 0.21, *p*<0.05) and social support (β = -0.20, *p*<0.05). 3) Patients reported higher anxiety when they had lower perceived social support (r = -0.32, p<0.001) and unsupportive social interactions (r = 0.41, p<0.001) 4) Patients reported higher depression when they had lower perceived social support (r = -0.33, p<0.001) and unsupportive social interactions (r = 0.44, p<0.001) |
| Mallinckordt et al., 2012 | Social support (SPS-M) | Psychological distress (BSI) | NR | 1) Patients demonstrated less psychological distress when they had higher social support (T1:r = -0.26, p<0.01; T2: r = -0.44,p<0.01). 2) A significant association was found between social support and psychological distress at both T1 and T2 (all p<0.01) |
| Popoola & Adewuya, 2012 | Perceived social support (indicating on Likert scale) | Depression (MINI) | NR | 1) Depression was associated with perceived social support (p = 0.001). 2) Perceived poor social support was a significant predictor of depression (*β* = 1.078, p = 0.014) |
| Aguado Loi et al., 2013 | Social support group attendance (Demographic questionnaire) Satisfaction with social support (Demographic questionnaire) | Depression (PHQ-9) | NR | 1) Increased depression was associated with satisfaction with family/peer support (β = -0.42, p<0.01). 2) The amount of support from family and friends was associated with depressive symptoms (β = -0.36, *p*<0.01). |
| So et al., 2013 | Social support (MOS-SSS) | Anxiety and Depression (HADS) | None | Patients showed lower levels of depression (β = -0.37, *p*<0.05) and anxiety (β = -0.28, *p*<0.05) when they had greater social support |
| Waters et al., 2013 | Perceived social support (MOS-SSS) | Emotional well-being (RAND 36-Item Health Survey) Worry (FACT-B) | Demographics, depression history and trait anxiety, cancer stage, types of surgical and adjuvant treatments | 1) Patients showed higher worrying about cancer progression when they had lower social support (r = 0.16, p<0.05). 2) Patients with higher social support reported better emotional well-being (Wilks' Λ = l0.86, F[24,1,320] = 2.9, p < .0001) |

(*Continued*)

**Table 3.** (*Continued*)

| Author (year) | Social relationship (measures) | Affective symptoms (measures) | Analysis adjusts for | Key findings |
|---|---|---|---|---|
| Yi & Kim, 2013 | Social support (PRQ-II) | Depressive symptom (CES-D) | NR | Patients with low social support reported higher depression (r = -.585, $p < .0001$). |
| Boinon et al., 2014 | Perceived social support (SSQ6) | Psychological distress (Impact of Event Scale) Depressive symptoms (Beck Depression Inventory) | NR | 1) A higher level of depressive symptoms at T2 (after adjuvant therapy) was associated with lower quantity of support (r = -.20, $p<0.05$), instrumental support (r = -0.26, $p<0.01$), and informational support at T1 (before adjuvant therapy) (r = -0.20, $p<0.05$) 2) Patients who perceived a higher instrumental support at T1 reported a lower level of depressive symptoms (β = -0.27, $p<0.05$) at T2 |
| Hasson-Ohayon et al., 2014 | Social support (CPASS) | Psychological distress (BSI) | None | 1) A significant association was found between social support and depression in younger patient group (β = -0.32, p = 0.016). 2) No significant association was found between social support and depression in older patient group. |
| Hughes et al., 2014 | Social support (ESSI) | Depression (CES-D) | Demographics, comorbidities, cancer stage, time since treatment | Patients with lower social support at T1(prior to cancer treatments) experienced higher level of depressive symptoms from T1 to T2 (6 months after the completion of cancer treatments) (β = -.47, t(137) = -2.97, p = 0.004) than patients with more social support. |
| Schleife et al., 2014 | Social support (VAS) | Anxiety and Depression (HADS) | NR | 1) Patients receiving social support showed less depression (r = -0.43, p<0.01) as well as anxiety (r = -0.36, p<0.01). 2) Higher social support decreased mental distress (β = -0.37, p<0.01). |
| Wang et al., 2014 | Social support (SSRS) Perceived social support (PSSS) | Depression (CES-D) Anxiety (STAI) | NR | Patients with strong perceived social support (β = -0.29, $p<0.01$; β = -0.23, $p<0.01$) and objective social support (β = -0.12, $p<0.05$; β = -0.14, $p<0.05$) reported lower depression as well as anxiety |
| Borstelmann et al., 2015 | Perceived social support (MOS-SSS) Marital subscale of Perceive partner support (CARES) | Anxiety (HADS) | NR | 1) Unsupported/partnered patients had higher anxiety (p<0.0001) 2) Patients with lower social support (OR = 0.96, 95% CI = 0.95–0.97) and unsupported/partnered (OR = 2.09, 95% CI = 1.34–3.24) reported higher anxiety |
| Ozkaraman et al., 2015 | Social support (CPSSS) | Social image anxiety (SIAS) | NR | 1) Patients demonstrated higher anxiety about body image when they received support from the spouse and/or children, but it was lower among women who had support only from friends (KW = 16.20; p = 0.02) 2) Higher anxiety was associated with decreasing reliance support (r = -0.35, p<0.001) |
| Alfonsson et al., 2016 | Lack of social support (Self-report Questionnaire) | Anxiety and Depression (HADS) | NR | 1) Lack of social support at T1 (shortly after diagnosis) predicted anxiety at T1 (p<0.001). 2) Lack of social support at T1 and T2 predicted anxiety at T2 (3 years after diagnosis) (p = 0.027; p = 0.020). 3) Lack of social support at T1 predicted depression at T1 (p = 0.004). 4) Lack of social support at T1 and T2 predicted depression at T2 (p = 0.01; p = 0.002). |
| Malicka et al., 2016 | Social support (BSSS) | Anxiety (STAI) Depression (BDI) | NR | No association was found between social support and anxiety as well as depression. |
| Berhili et al., 2017 | Family support (ask direct question about family assistance) | Anxiety and depression (HADS) | Demographics, taking analgesic and/or anxiolytic treatment, current treatment type | Patients demonstrated psychological distress when they had lack of social family support (p<0.001) |

(*Continued*)

**Table 3.** (Continued)

| Author (year) | Social relationship (measures) | Affective symptoms (measures) | Analysis adjusts for | Key findings |
|---|---|---|---|---|
| Fong et al., 2017 | Social Support (MOS-SSS) | Depressive symptoms (CES-D) Stress (Perceived Stress Scale) Positive and Negative Affect (Positive and Negative Affect Schedule) | Demographics, cancer stage | 1) Decline in social support quality predicted increase in depression (p = 0.003), negative affect (p = 0.05), and stress (β = -0.22, p = 0.01). 2) Decreases in social support quantity (β = -0.20) predicted increases in stress. 3) No association was found between social support quantity and negative and positive affect. |
| Moon et al., 2017 | Receiving emotional/ instrumental support (counting the total number of message containing emotional/instrumental support expression) | Depression (CES-D) | Demographics, total volume of message | 1) Patients who received emotional support by cancer survivors demonstrated greater reduction of depression (β = -0.32, $p<0.001$). 2) No association was found between receiving emotional support provided by other new patients and the reduction in depression. |
| Schellekens et al., 2017 | Social support (MOS-SSS) | Mood disturbances (POMS) | NR | In MBCR group, patients with increased social support reported changes in their mood disturbances (β = -0.24, p = 0.004) |
| Su et al., 2017 | Family support (APGAR) | Psychiatric diagnosis (MINI) | NR | Higher family support was associated with lower risk for major depressive disorder (β = 0.87, $p<0.05$). |
| Thompson et al., 2017 | Social support (MOS-SSS) | Depressive symptoms (CES-D) | Randomization assignment, levels of general health, depressive symptoms at baseline | 1) Patients with lower initial levels of social support demonstrated more severe depressive symptoms (β = 0.33, $p<0.001$) 2) Patients with lower baseline social support (β = -0.20, $p<0.05$) as well as greater decline in social support (β = -0.40, $p<0.05$) over time demonstrated more depressive symptoms over time |
| Tomita et al., 2017 | Social support (MOS-SSS) | Depressive symptoms (CES-D) | NR | Higher perceived social support decreased depressive symptoms (β = -0.25) |
| Bright & Stanton, 2018 | Social support (ISEL-12) | Depression (CES-D) | Demographics, medical factors, number of children | Greater social support at baseline was associated with lower depressive symptoms at 1month after hormone therapy (β = -0.41, $p<0.001$) |
| Schmidt et al., 2018 | Perceived social support (MSPSS) | Affective fatigue (FAQ) | Socio-demographics, clinical characteristics | Patients with poor social support (p = 0.001) demonstrated increased affective fatigue |
| Escalera et al., 2019 | Social support (MOS-SSS) | Psychological distress (BSI-18) | Demographics, time since diagnosis, adjuvant breast cancer treatment, cancer stage, history of depression | 1) Patients demonstrated fewer depressive symptoms when they had emotional/informational support (β = -0.17, p = 0.01), tangle support (β = -0.12, p = 0.03), positive social interaction (β = -0.13, p = 0.03) 2) Tangible support (β = -0.16, p = 0.006), affectionate support (β = -0.21, p = 0.001), and positive social interaction (β = -0.14, p = 0.02) were negatively associated with anxiety symptoms |
| Wondimagegnehu et al., 2019 | Social support (MSPSS) | Depression (PHQ-9) | NR | Depressed patients were found to have lower social support than non-depressed women (p = 0.027) |
| Janowski et al., 2020 | Social support (Disease-related Social Support Scale) | Depression (BDI) | NR | 1) Women with greater social support demonstrated lower depression than those with lower social support (t = 4.08, p<0.001) 2) Spiritual support was a significant predictor of depressive symptoms ($R^2$ = 0.27, β = -0.52, t = -5.01, p<0.001). |
| Debretsova &Derakshan., 2021 | Social support (MOS-SSS) | Anxiety and depression (HADS) | NR | Patients with greater social support demonstrated lower depression (r = -0.50, p<0.001) |
| Fisher et al., 2021 | Social support (MOS-SSS) | Depression (CES-D) | Demographics, medical factors | Patients with greater social support demonstrated lower depression (emotional support, β = -3.17, p<0.001) |

(*Continued*)

**Table 3.** (*Continued*)

| Author (year) | Social relationship (measures) | Affective symptoms (measures) | Analysis adjusts for | Key findings |
|---|---|---|---|---|
| Zamanian et al.,2021 | Social support (MOS-SSS) | Anxiety (DASS-A) and depression (DASS-D) | Demographics, medical factors, spouse's education, house mates | Patients with greater social support demonstrated lower anxiety and depression (r = -0.26~-0.38, p<0.001) |
| Okati-Aliabad et al., 2022 | Social support (MSPSS) | Anxiety and depression (HADS) | NR | Patients with greater social support demonstrated lower depression (r = -0.21,p<0.001) |
| **Social support and /or social constraints** | | | | |
| Schmidt & Andrykowski 2004 | Social support (DUKE-SSQ) Social constraints (SCS) | Anxiety and Depression (HADS) | NRv | 1) Patients with greater social support demonstrated lower depression (β = -0.23, p<0.001) 2) Patients with greater social constraints demonstrated greater depression as well as anxiety (β = 0.31, p<0.001; β = 0.34, p<0.001) |
| Wong et al., 2018 | Social constraints (Social constraints scale) Social support (Chinese version of MOS-SSS) | Depressive symptoms (CES-D) | Demographic, medical variables, cancer stage | 1) The indirect effect of social constraints on depressive symptoms through social support was significant (β = 0.11, p<0.01) 2) The direct effect of social support on depressive symptoms was significant (β = -0.28, p<0.01) |
| Lally et al., 2019 | Social constraints | Depressive symptoms (CES-D) | No covariates | 1) Patients who perceived social constraints from family/friends and spouse/partner reported higher depressive symptoms 2) Patients who experienced increased on family/ friends social constraints reported no changes in their depressive symptoms (p = 0.049) 3) Patients who experienced decreased family/friends social constraints reported decreased depressive symptoms (p = 0.049) |
| **Social support and family functioning (family conflict and family stress)** | | | | |
| Lueboonthavatchai, 2007 | Social support (SSQ) Family functioning (Family relationship and functioning questionnaire) | Anxiety and depression (HADS) | NR | 1) Patients' anxiety and depression were associated with social support (p<0.001) and family relationship and functioning (p<0.001). 2) Poor family relationship and functioning was a predictor of anxiety and depression (p<0.05). |
| Mantani et al., 2007 | Family functioning (FAD) | Anxiety (Zung self-rating anxiety scale) Depression (Zung self-rating depression scale) | NR | Patients demonstrated higher depression when they perceived inappropriate affective responsiveness among family members (β = 0.59, p<0.01). |
| Ashing-Giwa et al., 2013 | Social support (MOS-social support survey) Family stress (five-items from Life Stress Scale) | Depressive symptom (CES-D) | NR | Patients with low social support (r = -.37, p < .01) as well as family stress (r = .522, p = < .01) reported more depressive symptoms |
| Segrin et al., 2018 | Family conflict (Family Assessment Device) | Anxiety (PROMIS-Anxiety short form) Depressive symptoms (CES-D) | NR | 1) Patients demonstrated higher depressive symptoms when family conflict was high (β = 0.17, p<0.01) 2) Patients demonstrated higher anxiety when their family conflict was high (β = 0.11, p<0.05) |
| **Quality of relationships** | | | | |
| Giese-Davis & Hermanson, 2000 | Quality of couple's relationship (FRI): cohesion, expression, conflict | Mood disturbance (POMS) | Income | Patients demonstrated lower mood disturbance when they rated the relationship (w/partners) greater in cohesion-expression (β = -0.42, p<0.01) as well as greater in conflict (β = -0.40, p<0.001) |
| Manne et al., 2007 | Relationship satisfaction (DAS) | Psychological distress (Mental Health Inventory) | Sociodemographic, ECOG, surgery type, functional impairment, time since diagnosis, length of relationship | Greater patient relationship satisfaction was associated with decreased patients' psychological distress (β = -0.07, p<0.0001) |

(*Continued*)

**Table 3.** (Continued)

| Author (year) | Social relationship (measures) | Affective symptoms (measures) | Analysis adjusts for | Key findings |
|---|---|---|---|---|
| Segrin et al., 2007 | Relationship satisfaction (RAS) | Anxiety (PANAS, SF-12, ICS, and GSDS) Depression (CES-D) | NR | 1) No association was found between patients' anxiety and her reported relationship quality<br>2) Higher anxiety was found in patients when their partners reported dissatisfied relationship quality (T1: r = -0.20, p<0.05; T2:r = -0.28, p<0.01; T3:r = -0.27, p<0.05) |
| Al-Zaben et al., 2015 | Marital quality (SPS&QMI) | Anxiety and Depression (HADS) | NR | No significant association was found of anxiety/depression with the quality of the marital relationship |
| **Structural Aspect of Social Relationships** | | | | |
| Simpson et al., 2002 | Social Integration (ISSSI) | Mental Health (SCL&SCID) Depression (BDI) | Age, group membership, GAF, BDI, and GSI scores, baseline social support score | 1) Women who had psychiatric illness assessed by SCID had lower social support (p<0.001).<br>2) Social integration was not a predictor of the present of psychiatric illness.<br>3) Social integration (adequacy of close relationships) was a predictor of depression at 1-year post intervention (β = -0.23, p<0.01).<br>4) Social integration (adequacy of more distant supports) was a predictor of global severity of depression at 1 year post-intervention (β = -0.36, p<0.001). |
| **Both Aspects of Social Relationships** | | | | |
| Brothers & Andersen, 2009 | Perceived social support (PSS-F) Social network index (SNI) Presence of significant other/romantic partner | Depression (CES-D) | Physical functioning | 1) Depression was not associated with perceived social support<br>2) Patients' depression at both initial and follow-up was associated with the presence of support person (r = -0.25, p<0.05; r = -0.44, p<0.05)<br>3) The presence of significant others (β = -0.26, p<0.01) was a significant predictor of depression at follow-up. |
| Gagliardi et al., 2009 | Social network (Social Network List) Social support (1 to 4 Likert-type scale) | Anxiety (ASQ) Depression (CDQ) | NR | 1) Patients demonstrated lower anxiety (r = -0.43, p<0.01) and depression (r = -0.35, p<0.05) when they had strong informational support from kins<br>2) Patients demonstrated lower anxiety when they had strong emotional support (r = -0.356, p<0.05) from kins<br>3) No association was found between social network and patients' anxiety and depression |
| Puigpinos-Riera et al., 2018 | Social network (SNI) Social support (MOS-SS) Co-habitation at home | Anxiety and depression (HADS) | NR | 1) High risks of depression and anxiety were associated with social isolation (p = 0.00; p = 0.00) and low social support (p = 0.00; p = 0.00)<br>2) Living alone was associated with anxiety (p = 0.011). |
| Wang et al., 2019 | Social support (MOS-SSS) Social network index (count a total number of people who talk at least once every two weeks) | Depression and anxiety (PROMIS-short form) | Demographics, the level of acculturation (only for Chinese women), and clinical variables | 1) Patients showed more depression and anxiety when they had less social support (all p<0.05)<br>2) No association was found between social network and patients' anxiety and depression |

(*Continued*)

**Table 3.** (Continued)

| Author (year) | Social relationship (measures) | Affective symptoms (measures) | Analysis adjusts for | Key findings |
|---|---|---|---|---|
| Liu et al., 2021 | Social support (Social support rating scale) Social network (social isolation subscale of Lubben's social network) | Anxiety and depression (HADS) | NR | 1) Patients with lower social support (r = -0.334, p<0.01) and greater social isolation (r = 0.369, p<0.01) demonstrated greater anxiety. 2) Patients with lower social support (r = -0.289, p<0.01) and greater social isolation (r = 0.466, p<0.01) demonstrated greater depression. |

MOS-SSS = Medical Outcomes Survey-Social Support Survey; CARES = Cancer Rehabilitation Evaluation System; HADS = Hospital Anxiety and Depression Scale; SSS = Social Support Scale; CES-D = Center for Epidemiological Studies Depression Scale; PRQ-II = Personal Resource Questionnaire II; FACT-B = Functional Assessment of Cancer Therapy-Breast; POMS = Profile of Mood States; POMS-SF = Profile of Mood States-Short Form; NSSQ = Norbeck Social Support Questionnaire; SSQ6 = Social Support Questionnaire Short Form; STAI = State-Trait Anxiety Inventory; SSRS = Social Support Rating Scale; PSSS = Perceived Social Support Scale; FSSS = Familiar Social Support Scale; SSDS = Siddiqui-Shah Depression Scale; SCS = Social Constraints Scale; ASQ = Anxiety Scale Questionnaire; CDQ = Clinical Depression Questionnaire; MBSR = Mindfulness-Based Stress Reduction; SET = Supportive Expressive Group Therapy; CSOSI = Calgary Symptom of Stress Inventory; PANAS = Positive and Negative Affect Schedule; RAS = Relationship Assessment Scale; MIS = Lewis Mutuality and Interpersonal Sensitivity Scale; FHI = Family Hardiness Index; MDSS = Multi-Dimensional Support Scale; BDI-SF = Beck Depression Inventory-Short Form; DUKE-SSQ = Duke-UNC Functional Social Support Questionnaire; SCS = Social Constraints Scale; FRI = Family Relationship Index; MSPSS = Multidimensional Scale of Perceived Social Support; PHQ-9 = Patient Health Questionnaire 9; DAS = Dyadic Adjustment Scale; FACT-B: Functional Assessment of Cancer Therapy-Breast; BSI = Brief Symptom Inventory; CPASS = Cancer Perceived Agents of Social Support; ESSI = ENRICHD Social Support Instrument; ISEL = Support Evaluation List; SCL-90R: Standard Checklist-90-Revised; GSI = Global Severity Index; FAQ = Fatigue Assessment Questionnaire; MINI = Mini International Neuropsychiatric Interview; APGRA = Adaptability, Partnership, Growth, Affection, and Resolve; SNI = Berkman-Syme Network Index; FES = Family Environment Scale; PROMIS = Patient Reported Outcome Measurement Information System; CPSSS = Cancer Patient's Social Support Scale; SIAS = Social Image Anxiety Scale; MIHT = Multidimensional Inventory of Hypochondriacal Traits; USII = Unsupportive Social Interactions Inventory; SSQ = Social Support Questionnaire; TMD = Total Mood Disturbance; FACIT-G = Functional Assessment of Chronic Illness Therapy-General; SPS = Spousal Perception Scale; QMI = Quality of Marriage Index; ISEL-12 = Interpersonal Support Evaluation List (12 items); SPS-M: Social Provision Scale-Modified; BSSS = Berlin Social Support Scale; SCID = Structured Clinical Interview for DSM-III-R; VAS = Visual Analogue Scales; ISSB = Inventory of Socially Supportive Behaviors; MISSB = Modified Inventory of Socially Supportive Behaviors; SF-12 = 12-item Short Form Survey; ICS = Index of Clinical Stress; GSDS = General Symptom Distress Scale

received stronger emotional/subjective support, their experience of affective symptoms decreased. Some longitudinal studies showed that emotional/subjective support can function as a predictor of patients' anxiety and depression [34, 61, 62]. Additionally, improvements in affective symptoms occurred when tangible support such as material support/assistance (e.g., brochures) was provided [34, 37, 56, 59, 61].

In six studies, patients' affective symptoms were affected by source and satisfaction of social support received. When patients received support from their family members, including a spouse or children, they reported less anxiety and depression [19, 42, 63–65]. However, one study showed less depression and anxiety when support was received from friends compared to support from family [66]. In addition, higher satisfaction with support received was associated with the lower levels of patients' anxiety and depression [32, 63, 67–70]. A study reported that patients showed less affective symptoms when they were more satisfied with support from family than from friends [66]. Patients' affective symptoms were not related to whether they were satisfied with their friend's support but were related to the amount of support received from a friend.

**Social support and/or social constraints.** Three studies have examined the association of patients' depression with social constraints [71–73]. Patients who perceived social constraints from family (including spouse/partners) or friends showed higher depressive symptoms. However, patients showed lower depression when they had decreased family/friend social constraints. Patients reported no change in depression when social constraints increased [72, 73]. Also, lower depression was reported when patients received greater social support.

**Social support and family functioning.**   One study found that both patients' anxiety and depression decreased when they had greater social support and a better-functioning family [74]. Furthermore, family functioning predicted the levels of patients' anxiety and depression [74]. In line with this finding, three other studies also found higher depression in patients when they perceived poor/ineffective family functioning. Specifically, depression greatly increased when patients experienced inappropriate responses from family [75], conflicts between its members (i.e., family conflict) [76], and stress due to the demands on the family (i.e., family stress) [77].

**Quality of relationships.**   Four studies investigated the quality of relationships with patients' partners/spouses that patients perceived and assessed its association with their affective symptoms. Of those four, two of them failed to show any significant associations of affective symptoms with the quality of couple/marital relationships [78, 79]. However, one study showed that anxiety was not associated with patient's reported relationship quality but with the partner's reported relationship quality [79]. The other two studies showed that patients' psychological distress and mood disturbance increased when patients reported unsatisfying relationships with their spouse/partners [80, 81]. Specifically, one study found that lower mood disturbance was reported when patients have a partner relationship with greater cohesion and expression (i.e., open communication) as well as more constructive conflicts [81]. The authors interpreted constructive conflicts as an indicator of greater engagement in the relationship with partners. In other words, constructive conflicts can occur due to greater discussion/understanding of each other's specific needs, and this constructive conflict can help reduce patients' mood disturbances.

## Structural aspects of social relationships

Structural aspects of social relationships refers to the structure of social networks, such as the size and the linkage between members within a social network [9]. This review included social integration as the structural social relationships (Table 3).

**Social integration.**   In this review, one RCT that investigated the effect of psychoeducational intervention examined the association between social integration and affective symptoms of breast cancer patients [82]. Social integration did not show any associations with the presence of psychiatric illness. However, one year after psychoeducational intervention, patients showed overall less depression when they perceived adequacy of both close relationships and more distant social ties (i.e., greater social integration).

## Both aspects of social relationships

**Social networks.**   Social networks can be assessed through whether individuals have important persons in their lives, type (e.g., friends or family) and duration of the relationship, and the frequency of contact with that persons [83]. Five studies assessed both of social networks and social support and their associations with patients' affective symptoms. Patients showed less anxiety and depression when they had stronger social support and social network (i.e., lower social isolation) [84–88]. Specifically, one longitudinal study showed that patients' depression at both initial and follow-up appointments was improved when they had a support person [84]. Another study also reported that living alone (without having a support person) contributed to increased anxiety as well as depression levels [86, 88].

## Discussion

Seventy studies met the inclusion criteria and informed this review. None of the included studies examined the association of social relationships with patients' cognitive symptoms, thus

including studies that investigated the association of social relationships with affective symptoms. Of those 70 studies, four studies completely failed to show significant associations of affective symptoms with any aspects of social relationships [22–24, 78]; we found that most patients who participated in those four studies were primarily treated with surgery, which could be interpreted as showing very early stage breast cancer. However, in patients with advanced cancer (metastatic disease), better social relationships are associated with lower levels of their affective symptoms [19, 20, 67, 81]. This finding suggests that patients with advanced stage cancer can benefit from social relationships in managing their affective symptoms compared with those with early stages of cancer.

This review found that the level of social support and its association with affective symptoms change throughout the cancer treatment trajectory. Patients reported a decrease in social support before and after cancer treatments [14, 61]. Specifically, a continuous decrease in emotional support was found after surgery for breast cancer, whereas informational or tangible support increased right after surgery and then dropped over time [34, 63]. Although the overall levels of social support showed a decreased trend, the magnitude of its association with affective symptoms increased over time [14, 29, 61]. These findings suggest the need to assess the level of social support and implement programs to optimize social support, especially for those at the end of cancer treatments.

We further found several factors that help explain the link between social support and affective symptoms. A study showed that women with higher social support appraised their illness as less stressful situations and, in turn, fewer mood disturbances [61]. Another study also found that patients who perceived lower level of social support tend to choose passive coping strategy (i.e., self-blame) rather than active coping (i.e., positive reframing), which in turn decrease emotional well-being [54]. In contrast, those who received higher level of social support are more likely to rely on active coping, resulting in enhancing emotional well-being [54]. Similarly, Hills and colleagues (2011) reported higher self-blame and lower social support predict greater levels of depression and anxiety [55]. In addition to the appraisal of illness and coping strategies, several other studies showed that demographic information (age, income, education, marital status), clinical information (cancer stage, type of surgery, treatment types), physical function, and coping styles have comparable effects to social support on mood disturbances [37, 77]. Further studies are needed to identify factors that explain the link between social support and affective symptoms; doing so will help develop targeted interventions.

In this review, social relationships were divided into their function and structural aspects. Functional aspects of social relationships include four variables: social support, social constraints, family functioning, and quality of relationship. Social support refers to aid provided (e.g., emotional or instrumental) through contact with one's social networks (e.g., friends or family) [9, 49, 83], whereas social constraints are social conditions that hinder individuals' expression of stressors due to unsupportive, misunderstood, or isolated responses from others [89]. Our findings clearly show that patients' affective symptoms are positively associated with the quantity (e.g., time spent or availability), type (e.g., tangible aid or empathy), source (i.e., who provided support), and satisfaction from the support that they received. Additionally, greater levels of affective symptoms are associated with negative social interactions (i.e., social constraints) and poor family functioning.

Compared with three functional aspects above, findings regarding the quality of social relationships are not consistent. Some studies reported that the quality of relationships is associated with patients' affective symptoms [80, 81], but others do not [78, 79]. Furthermore, one study showed that patients who reported greater conflicts in relationships with partners also reported lower mood disturbances [81].This inconsistency can occur due to differences in sample characteristics. For example, patients included in Giese-Davis and Hermanson (2000)'s

study reported more metastatic diseases compared to those of other three studies [78–80]. In addition, the study conducted by Al-Zaben and colleagues (2015) included married couples and investigated their marital quality, whereas other three studies focused on the relationship quality/satisfaction with their significant others. Future studies would benefit from ensuring consistency and specificity in defining and measuring quality of social relationships.

Similarly, structural aspects of social relationships also show an association with patients' affective symptoms. All six included studies showed that having a support person, not living alone, and building close relationships with others are factors that lower patients' affective symptoms. It is possible that patients with larger social networks and greater social integration may increase the odds that patients will have friends and family who survive as peer and familiar support [90]. This support can be beneficial while patients are managing symptoms from disease and/or treatment [90]. Additional research is needed to understand how structural and functional aspects of social relationships interact with biological factors (e.g., cytokines, HPA axis dysfunction) to influence patients' affective symptoms. This understanding may help identify important concepts for models that promote social relationships in breast cancer patients that will help improve their affective symptoms.

### Implication for practice

Most interventions for those with affective symptoms have primarily focused on managing their internal clinical characteristics. However, our findings reveal that positive social relationships benefit in mitigating affective symptoms of women with breast cancer. Thus, healthcare providers need to educate patients about the importance of building positive social relationships and encourage them to participate in a supportive network of friends and family members. Specifically, patients with advanced cancer (i.e., metastatic status) may find it highly beneficial to have access to support groups that are relevant to their specific needs. For example, health professionals can encourage them to participate in interventions that include components such as communication skills training or coordinating coping responses; in turn, this will help improve the quality of relationships.

Based on findings from this review paper, there is a need to capitalize on existing relationships that patients perceive as beneficial in their everyday lives like those considered as family members. Family intervention development that aims to lower affective symptoms as well as improve quality of life and well-being may be a suitable next step in improving patient outcomes. For example, family-based group tasks that improve family functioning or family conflicts can be provided to breast cancer patients and their family members as a part of the intervention for improving affective-cognitive symptoms.

Lastly, it is important in the clinical setting to assess social support and social constraints. This type of assessment may be helpful in preventing, and furthermore, mitigating their affective symptoms. Assessment tools for social relationships including social support or social constraints can be built into the medical chart to alert for clinical staff to address. Additionally, establishing a social system to support coordination of various types of social relationships from healthcare professionals may yield positive affective outcomes in breast cancer patients. While evidence supporting this association is limited, more studies on the impact of social relationships on affective symptoms of breast cancer patients are recommended.

### Limitations

Our study goal was to find literature that examined the association between social relationships and cognitive symptoms among breast cancer patients. The review of literature yielded that there are no published studies that study this association based on our review criteria. During

the literature search, we found several studies that investigated this association in healthy older adults [10, 11]. However, no studies have been conducted in the context of breast cancer. We only included articles that explore the association between social relationships and affective symptoms of breast cancer patients. Future research that considers the effect of social relationships on cognitive symptoms in breast cancer is needed to advance our knowledge in cancer symptom science.

Approximately half of the included studies did not report confounding factors (e.g., sociodemographic) and did not adjust for these factors. This is an important limitation because the associations between social relationships and a patients' affective symptoms could differ depending on confounding factors. Thus, it is essential to report and adjust for confounding factors using statistical methods.

Another limitation is that fewer included studies focused on assessing the association between structural aspects of social relationship and patients' affective symptoms. To fully understand the role of structural aspects of social relationships on patient's affective symptoms, further studies are needed that include diverse aspects of social relationships are needed.

Lastly, we found that most included studies RCTs design. To better understand the influence of social relationships on patients' affective symptoms, studies with observational longitudinal studies are needed. Additionally, most patients included this study were White, which could impede generalizability of the study findings. Therefore, a large and heterogeneous sample is needed for future studies to be representative of all women breast cancer patients from all ethnicities.

## Conclusions

This scoping review summarized current evidence concerning social relationships that are associated with affective symptoms of a breast cancer patient. Of the identified social relationships, social support was most identified, followed by social constraints, family functioning, quality of relationships, social networks, and social integration. Our review results support the concept of an association between social relationships and affective symptoms of breast cancer patients, although the specific nature of this association remains unclear. Understanding different aspects of social relationships and their differential effects on patients' affective symptoms will contribute to development of interventions for best practices to support the well-being of this patient population.

## Supporting information

**S1 Table. Search strategy.**
(DOCX)

**S1 Checklist. PRISMA-ScR checklist.**
(DOCX)

## Author Contributions

**Conceptualization:** Yesol Yang, Timiya S. Nolan.

**Data curation:** Yesol Yang, Yufen Lin, Grace Oforiwa Sikapokoo, Se Hee Min, Nicole Caviness-Ashe, Jing Zhang, Leila Ledbetter.

**Formal analysis:** Yesol Yang, Yufen Lin, Leila Ledbetter.

**Investigation:** Yesol Yang, Yufen Lin, Leila Ledbetter, Timiya S. Nolan.

**Methodology:** Yesol Yang, Yufen Lin, Grace Oforiwa Sikapokoo, Leila Ledbetter.

**Project administration:** Yesol Yang.

**Supervision:** Yesol Yang.

**Validation:** Yesol Yang, Timiya S. Nolan.

**Writing – original draft:** Yesol Yang.

**Writing – review & editing:** Yesol Yang, Yufen Lin, Grace Oforiwa Sikapokoo, Se Hee Min, Nicole Caviness-Ashe, Jing Zhang, Leila Ledbetter, Timiya S. Nolan.

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
