## [Decision Letter · Decision Letter 0]

23 May 2022

PONE-D-22-04918Social relationships and their associations with affective symptoms of women with breast cancer: A scoping reviewPLOS ONE

Dear Dr. Yang,

Thank you for submitting your manuscript to PLOS ONE. After careful consideration, we feel that it has merit but does not fully meet PLOS ONE’s publication criteria as it currently stands. Therefore, we invite you to submit a revised version of the manuscript that addresses the points raised during the review process. I have received one review from an expert in the field, who has provided very detailed feedback. I have also independently reviewed your manuscript. The reviewer and I found your paper very interesting and a topic worthy of review. However, we both have major concerns that prevent publication in its current form. I will not reiterate all of the reviewer's points, but I do want to emphasize my agreement with the suggestion to move discussion of cognitive symptoms to the discussion/limitations section. Below are additional concerns and suggestions I have: 1. Expand the introduction to clearly explain/define what is meant by affective symptoms. Throughout the paper, depression and anxiety are primarily discussed, but other affective symptoms and measures are included in the review. Clear operationalization and justification of what does and does not constitute affective symptoms is necessary. It would also help to better situate this review in the existing literature. Why is important to examine affective symptoms in breast cancer populations? 2. Similarly, the discussion of social relationships in the introduction should be expanded. A more representative discussion of the literature linking social relationships to affective symptoms, particularly your definition of affective symptoms, is necessary. In your review, you emphasize the distinction between functional and structural components of social relationships. Why is this distinction important? What evidence is there to suggest that certain components of social relationships are more important than others? To prepare the reader for the review, it would be helpful to introduce and define different components of social relationships (e.g., social integration, social support, functioning), particularly those components covered in the review. 3. In the inclusion/exclusion criteria, provide detailed operational definitions for affective symptoms and social relationships. What measures were acceptable versus not? As is, the review covers a broad range of affective symptoms and social relationship measures, with some variables or concepts only represented once. It may be worth refining your operational definitions/inclusion criteria to have a more focused review with clearer implications. 4. I take it articles were limited to peer-reviewed papers. Please specify. 5. Did any of the papers involving secondary data analysis use datasets from any of the other articles included in the review? Is there independence of findings? 6. In the discussion, it would be helpful to elaborate on potential methodological or sample differences that might explain inconsistent findings. 7. The practical implications section is pretty generic. I recommend developing more nuanced suggestions based on the findings of the review and highlighting unique components of social relationships. 8. Throughout the paper, causal language is used regarding the relationship between social relationships and affective symptoms. As most of the literature reviewed is correlational, causal inference is not possible. Tighten up the language to more accurately discuss the association.     

We look forward to receiving your revised manuscript.

Kind regards,

Natalie J. Shook

Academic Editor

PLOS ONE

Journal Requirements:

When submitting your revision, we need you to address these additional requirements. 1. Please ensure that your manuscript meets PLOS ONE's style requirements, including those for file naming. The PLOS ONE style templates can be found at https://journals.plos.org/plosone/s/file?id=wjVg/PLOSOne_formatting_sample_main_body.pdf and https://journals.plos.org/plosone/s/file?id=ba62/PLOSOne_formatting_sample_title_authors_affiliations.pdf 2. We note that you have stated that you will provide repository information for your data at acceptance. Should your manuscript be accepted for publication, we will hold it until you provide the relevant accession numbers or DOIs necessary to access your data. If you wish to make changes to your Data Availability statement, please describe these changes in your cover letter and we will update your Data Availability statement to reflect the information you provide 3. In your Data Availability statement, you have not specified where the minimal data set underlying the results described in your manuscript can be found. PLOS defines a study's minimal data set as the underlying data used to reach the conclusions drawn in the manuscript and any additional data required to replicate the reported study findings in their entirety. All PLOS journals require that the minimal data set be made fully available. For more information about our data policy, please see http://journals.plos.org/plosone/s/data-availability. "Upon re-submitting your revised manuscript, please upload your study’s minimal underlying data set as either Supporting Information files or to a stable, public repository and include the relevant URLs, DOIs, or accession numbers within your revised cover letter. For a list of acceptable repositories, please see http://journals.plos.org/plosone/s/data-availability#loc-recommended-repositories. Any potentially identifying patient information must be fully anonymized.

Reviewers' comments:

Reviewer's Responses to Questions

**Comments to the Author**

1. Is the manuscript technically sound, and do the data support the conclusions?

Reviewer #1: Partly

2. Has the statistical analysis been performed appropriately and rigorously? 

Reviewer #1: N/A

3. Have the authors made all data underlying the findings in their manuscript fully available?

Reviewer #1: Yes

4. Is the manuscript presented in an intelligible fashion and written in standard English?

Reviewer #1: Yes

5. Review Comments to the Author

Reviewer #1: This manuscript is about social relationships and their association with affective symptoms of women with breast cancer. The topic is important. This is a scoping review, and the authors addressed current evidence of associations of functional and structural social relationships with affective symptoms in breast cancer survivors using the PRISMA-ScR guideline. The main finding is that positive social relationships may contribute to mitigating affective symptoms of women with breast cancer, which suggests that education and participation in supportive networks are needed to manage affective symptoms. The reviewer has the following further comments.

1. In the Abstract (lines 62-64) section, the authors state “Of seventy included studies, none of them focused on cognitive symptoms of breast cancer patients; thus, in this review, we focused on only the affective symptoms of breast cancer patients and their association with patients’ both aspects of social relationships.” – this is not a main finding. The main point is that social support/network may be associated with better affective symptom outcomes. Please rewrite the result.

2. How about focusing on writing the association between social support and affective symptoms throughout and briefly mentioning social relationships with cognitive symptoms in the limitation section as no studies on cognitive symptoms were included?

3. It would be helpful to add a meta-analysis of the associations between social support and affective symptoms to estimate how strong an association is between two variables given a great volume of the studies included in the review.

4. The authors may include the study quality evaluation for each study in the table.

5. It seems that intervention articles (e.g., mindfulness) were also included in the analysis. Please clarify the inclusion criteria in the method section.

6. Social relationships with affective symptoms would differ depending on the cancer treatment trajectory (pre-, during, and post-treatment). Survivors would need more social relationships/support before or during cancer treatment than those after cancer treatment. It would be helpful to take a further look at how social relationships with affective symptoms may differ over the treatment trajectory.

7. The discussion section simply addressed the functional and structural social relationships. As the authors indicated in Table 3, several variables seem to influence the association between social relationships and affective symptoms, such as desirability, cancer stage, comorbidity, sociodemographics (e.g., income, race, employment), lifestyle, etc. It would be helpful to address how mediating or moderating factors play a role in affecting the influence of social support on affective symptoms. Also, the authors may add the association between social support/network and affective symptoms in other cancer types and the role of social support in addressing cancer health disparities.

Minor comments:

1. Please do not use the term "Caucasian." It is a perjorative term with connotations of racial supremacy. The correct term is "White."

2. Line 127: We reported the findings using…

3. Please change PRISMA-Sc to PRISMA-ScR.

4. Table 1: In the “exclusion criteria” section, please remove periods (.)

5. Line 229-230: “In six studies” was written twice.

6. Line 248: Please use either one (In contrast/However).

7. Please abbreviate terms upon first use – e.g., Randomized controlled trials (RCT)

8. Table 3: hose mates -> house mates

9. Please keep consistency between NR or “not reported” in the table.

10. An editor may help with the writing of the manuscript.

6. PLOS authors have the option to publish the peer review history of their article (what does this mean?). If published, this will include your full peer review and any attached files.

Reviewer #1: No

---

## [Author Response · Author response to Decision Letter 0]

15 Jun 2022

We uploaded a document of "responses to reviewers" into the "Attach Files" folder.

---

## [Decision Letter · Decision Letter 1]

25 Jul 2022

Social relationships and their associations with affective symptoms of women with breast cancer: A scoping review

PONE-D-22-04918R1

Dear Dr. Yang,

We’re pleased to inform you that your manuscript has been judged scientifically suitable for publication and will be formally accepted for publication once it meets all outstanding technical requirements.

Kind regards,

Natalie J. Shook

Academic Editor

PLOS ONE

Additional Editor Comments (optional):

Reviewers' comments:

Reviewer's Responses to Questions

**Comments to the Author**

1. If the authors have adequately addressed your comments raised in a previous round of review and you feel that this manuscript is now acceptable for publication, you may indicate that here to bypass the “Comments to the Author” section, enter your conflict of interest statement in the “Confidential to Editor” section, and submit your "Accept" recommendation.

Reviewer #1: All comments have been addressed

2. Is the manuscript technically sound, and do the data support the conclusions?

Reviewer #1: Yes

3. Has the statistical analysis been performed appropriately and rigorously? 

Reviewer #1: N/A

4. Have the authors made all data underlying the findings in their manuscript fully available?

Reviewer #1: Yes

5. Is the manuscript presented in an intelligible fashion and written in standard English?

Reviewer #1: Yes

6. Review Comments to the Author

Reviewer #1: Minor comments:

Page 46 Line 199: Please remove “is needed”

Page 47. Line 211-214: Could you clarify the sentences? “we found that most included studies RCTs design. To better understand the influence of social relationships on patients’ affective symptoms, studies with alternative designs such as longitudinal studies and RCTs are needed.” Most studies included in this analysis were designed as RCTs? If RCTs were a common design in this analysis, why are RCTs needed as alternative designs for future studies?

Page 44 Line 146-148: “patients included in Giese-Davis and Hermanson (2000)’s study

were metastatic disease compared with other three studies (78-80) included early stage of breast cancer(78-80).”: Could you please restate this sentence to clarify? Patients included in G-D and Hermanson's study reported more metastatic diseases compared to those of other three studies (78-80)...?

Page 47 Line 216: Please remove “women”

It would be better to check if there are any ambiguous sentences and grammatical errors throughout the manuscript before publication.

7. PLOS authors have the option to publish the peer review history of their article (what does this mean?). If published, this will include your full peer review and any attached files.

Reviewer #1: No

---

## [Editor Report · Acceptance letter]

28 Jul 2022

PONE-D-22-04918R1 

Social relationships and their associations with affective symptoms of women with breast cancer: A scoping review 

Dear Dr. Yang:

I'm pleased to inform you that your manuscript has been deemed suitable for publication in PLOS ONE. Congratulations! Your manuscript is now with our production department. 

Kind regards, 

on behalf of

Dr. Natalie J. Shook 

Academic Editor

PLOS ONE